# The Neural Correlations of Olfactory Associative Reward Memories in *Drosophila*

**DOI:** 10.3390/cells13201716

**Published:** 2024-10-17

**Authors:** Yu-Chun Lin, Tony Wu, Chia-Lin Wu

**Affiliations:** 1Graduate Institute of Biomedical Sciences, College of Medicine, Chang Gung University, Taoyuan 33302, Taiwan; lyc2001bs@gmail.com; 2Brain Research Center, National Tsing Hua University, Hsinchu 30013, Taiwan; 3Department of Neurology, New Taipei Municipal TuCheng Hospital, Chang Gung Memorial Hospital, New Taipei City 23652, Taiwan; tonywu@cgmh.org.tw; 4Department of Biochemistry, College of Medicine, Chang Gung University, Taoyuan 33302, Taiwan

**Keywords:** *Drosophila melanogaster*, neural circuits, brain, reward memories

## Abstract

Advancing treatment to resolve human cognitive disorders requires a comprehensive understanding of the molecular signaling pathways underlying learning and memory. While most organ systems evolved to maintain homeostasis, the brain developed the capacity to perceive and adapt to environmental stimuli through the continuous modification of interactions within a gene network functioning within a broader neural network. This distinctive characteristic enables significant neural plasticity, but complicates experimental investigations. A thorough examination of the mechanisms underlying behavioral plasticity must integrate multiple levels of biological organization, encompassing genetic pathways within individual neurons, interactions among neural networks providing feedback on gene expression, and observable phenotypic behaviors. Model organisms, such as *Drosophila melanogaster*, which possess more simple and manipulable nervous systems and genomes than mammals, facilitate such investigations. The evolutionary conservation of behavioral phenotypes and the associated genetics and neural systems indicates that insights gained from flies are pertinent to understanding human cognition. Rather than providing a comprehensive review of the entire field of *Drosophila* memory research, we focus on olfactory associative reward memories and their related neural circuitry in fly brains, with the objective of elucidating the underlying neural mechanisms, thereby advancing our understanding of brain mechanisms linked to cognitive systems.

## 1. Introduction

Similar to embryonic development, complex behaviors are influenced by selective pressure and exhibit a remarkable degree of conservation across various animal phyla. This genetic perspective offers a foundational rationale for initially employing simple model systems to identify the specific genes associated with behavioral phenomena, followed by the manipulation of the expression or function of these genes to elucidate the underlying molecular and neural mechanisms. However, the reductionist objective to understand behavioral traits presents significant challenges. In contrast to other organs that function as homeostatic systems to maintain bodily functions within relatively narrow parameters in the face of environmental fluctuations, the brain is a distinctive organ that has evolved to exhibit plasticity [1,2]. Specifically, the brain has developed to interpret external stimuli, and to adapt its structure and function in response to specific experiences. This plasticity involves ensembles of neural circuits and cellular mechanisms that encode new experiences and facilitate adaptive changes in an organism’s behavioral responses [3,4,5]. The functional roles of these individual gene products are interconnected through complex gene and neural networks that continuously interact in response to environmental stimuli or experimental interventions. Consequently, while the unique characteristics of the brain endow organisms with remarkable behavioral plasticity, they can complicate our efforts to understand the intricacies of behavioral complexity.

*Drosophila melanogaster* is among the most extensively studied invertebrates and serves as a highly effective genetic model organism. The wide array of behaviors displayed by *Drosophila* in their natural environment indicates that these organisms are capable of utilizing memory to inform their decision-making processes [6]. The significance of *Drosophila* as an animal model for research into learning and memory, the behavioral expressions of brain plasticity, has been well established [7,8]. Despite the divergent evolutionary trajectories of insects and mammals, both groups share a common bilaterian ancestor characterized by complex physiological traits. Consequently, *Drosophila* and mammals exhibit analogous cellular pathways that influence neural function, brain connectivity, and ultimately behavior [9,10]. Comparative studies across several species have shown that numerous molecular pathways governing cellular homeostasis and functionality are remarkably conserved between *Drosophila* and humans [8,11,12]. The *Drosophila* is distinguished by its relatively uncomplicated neural circuits compared with vertebrates, allowing for a more effective investigation of the mechanisms by which the brain produces behavioral responses, as well as the opportunity to analyze and dissect neural circuits in greater detail. The behaviors exhibited by *Drosophila* are robust and share similarities with those of mammals and other vertebrates, making them amenable to simple assay and quantification [8,12,13,14,15,16,17]. Moreover, research involving flies has a history spanning over a century, and the genetic tools developed for *Drosophila* provide a powerful framework for establishing causal relationships among genes, neural networks, and behavior [18,19,20,21,22]. Studies involving the genetic analyses of *Drosophila* have facilitated the screening of gene mutations that influence behavior, while sophisticated genetic investigation methodologies have enabled the functional alteration of clusters of phenotypically associated genes, allowing the distinction of their individual contributions to learning and memory processes, neural circuitry, and the biochemical pathways associated with brain plasticity [6,23,24,25,26,27,28,29].

Instead of comprehensively examining all known genes implicated in *Drosophila* learning and memory, this review instead aimed to provide an experimental introduction to *Drosophila* reward-related olfactory associative memories, subsequently integrating the predominantly investigated forms of olfactory associative memory with the *Drosophila* olfactory nervous system. Our focus in this study will culminate in an analysis of contemporary research concerning water- and sugar-reward memory, as well as the neural circuits associated with these types of memory. This review further emphasizes the mechanisms underlying the formation of olfactory associative reward memories and investigates the current understanding of the integration of internal states and odor processing within the fly brain.

## 2. Memory Formation in *Drosophila*

### 2.1. Experimental Approaches in Drosophila Olfactory Associative Learning

Olfactory classical conditioning is the most researched type of learning and memory in *Drosophila*. Research on olfactory learning in *Drosophila* has thus provided valuable insights into the neural processes involved in learning and memory. The most commonly used paradigm for olfactory memory assays in *Drosophila*, known as Pavlovian conditioning, involves linking an odor as a conditioned stimulus (CS) with either an aversive (electric shock punishment) or appetitive (sucrose- or water-reward) stimulus as an unconditioned stimulus (US) to flies [6,22,30,31,32,33,34,35,36]. The process of classical olfactory conditioning further involves the convergence of sensory signals from the odor (CS) and aversive or appetitive stimuli (US) through distinct molecular pathways. This integration occurs within the brain regions where the neural circuits for both stimuli converge.

Typically, this conditioning procedure entails exposing groups of flies to two distinct odors, a CS+ and a CS−, with only the CS+ being presented concurrently with the US. Numerous studies have described the appetitive olfactory conditioning paradigms extensively [37,38,39]. Herein, we illustrate an associative water- (or sucrose-) reward paradigm, in which water (or sucrose) serves as a form of reinforcement (Figure 1). Prior to conditioning, the flies were subjected to a period of water or food deprivation lasting between 16 and 24 h while housed in glass milk bottles containing filter paper with dried sucrose or water-soaked filter paper. The training tube in the teaching machine, called the T-maze, served as the positive conditioned stimulus (CS+) and further contained filter paper soaked with water (or sucrose). The conditioning procedure involved placing approximately 50–75 thirsty (or hungry) flies in the elevator section of a T-maze. Initially, the flies were moved to the CS− tube, where an odor was presented for 2 min without water (or sucrose) reinforcement. Subsequently, following a 1 min interval of exposure to clean air, the flies were returned to the elevator and directed to the CS+ tube, which contained water (or sucrose) reinforcement as well as a different odor, for another 2 min period. Memory was assessed at specified intervals following the training session by presenting the flies with a selection between two distinct odors. Flies that effectively formed memories opted for the CS+ odor. The performance index (PI) was determined by subtracting the number of flies in the CS− tube from the number in the CS+ tube, and then dividing this difference by the total number of flies. The flies were individually transferred from each tube into vials and immobilized by freezing to allow accurate counting. To minimize any potential bias related to odor perception, a single PI was derived from the average scores of two distinct experiments. In these experiments, separate cohorts of flies with identical genotypes were conditioned with different combinations of 4-methylcyclohexanol (MCH) and 3-octanol (OCT) odors as the CS+ and CS− stimuli.

### 2.2. Different Types of Olfactory Associative Memory

The process of memory formation in *Drosophila* can be delineated into three primary phases: acquisition, consolidation, and retrieval. Each of these phases is characterized by specific neural circuits, molecular mechanisms, and genetic components that collectively facilitate the memory process. Acquisition pertains to the initial stage of memory formation, during which the organism learns to associate a particular stimulus, known as the CS, with a US. Consolidation involves the process of enhancing the stability of memories and their resistance to interference. Finally, retrieval constitutes the concluding phase of memory formation, wherein the stored memory is accessed and expressed. Memories can be classified into distinct categories based on their duration, as well as the various forms of reinforcement they receive. Early studies have indicated the presence of two distinct types of memory following a single aversive training session in response to anesthesia: anesthesia-sensitive memory (ASM) and anesthesia-resistant memory (ARM), which can persist for up to one day following a single training session [40,41,42,43,44,45]. In addition, multiple training sessions induced distinct types of memory compared to single training sessions. When multiple training sessions are performed without spaced intervals, such as in massed training involving five to ten consecutive sessions, memory retention is enhanced, lasting approximately three days. However, this type of memory is not influenced by the protein synthesis inhibitor cycloheximide. Conversely, spaced training, which involves 5 to 10 sessions of training with a 15 min interval between each session, results in protein synthesis-dependent memory formation that can endure for at least one week [45,46,47,48]. In other words, repeated exposure enhances memory formation, with spaced repetitions proving to be the most effective. These experimental manipulations conducted in wild-type flies indicate that an initial, unstable memory undergoes a process of consolidation over time, transforming into durable, long-lasting memory, a phenomenon observed in various animal species, including humans.

Examination of mutant and transgenic flies further revealed genes involved in olfactory memory formation, dividing memory into four distinct phases: short-term memory (STM), ASM, ARM, and long-term memory (LTM) [1,22,32,45,49]. STM appeared immediately following training. In the case of Aplysia, STM sensitization was associated with an increase in Cyclic Adenosine Monophosphate (cAMP) levels in neurons responsible for behavioral responses [50]. Similarly, investigation of flies with mutations in *dunce* and *rutabaga* revealed the involvement of cAMP components in STM [32,51,52,53,54,55,56,57]. Studies on *rugose* mutants have further demonstrated the molecular dissociation between STM and consolidated memory in *Drosophila*. Mutations in the *rugose* gene result in impaired immediate memory in aversive olfactory conditioning, whereas LTM is unaffected, indicating a separation of memory phases at the molecular level [58]. Consequently, STM in flies is characterized by increased learning levels immediately post-training, with a subsequent decay within 60 min, whereas ARM manifests gradually, reaching stable levels within two hours post-training [42,43,45].

An alternative method involves the classification of memories based on the type of reinforcement that flies receive, particularly rewards and aversive stimuli. *Drosophila* can establish positive “reward” memories linked to stimuli that predict the presence of a reward, such as water in thirsty flies, or sucrose in hungry flies [59]. This form of memory acquisition is characterized by a positive valence and is initiated when stimuli are associated with the anticipation of a reward. Conversely, *Drosophila* can further establish aversive memories associated with stimuli that signal the termination of a reward or delivery of a shock. These memories are distinguished by a negative valence and are formed when stimuli anticipate the withdrawal of a reward or are associated with the electrical shock punishment [60,61]. It could be anticipated that biochemical processes and neural pathways involved in the creation of various memories will vary.

## 3. The Olfactory Nervous System

Olfactory information related to the conditioned stimulus is transmitted to mushroom bodies (MB) by neurons located upstream of the olfactory nervous system. The olfactory nervous system of *Drosophila melanogaster* exhibits a high degree of structural and functional similarity to that in vertebrates, indicating that the fundamental principles elucidated in this context are likely applicable across different animal phyla [62]. Within the olfactory system of *Drosophila*, the neural pathway responsible for conveying olfactory information from the antennal lobes to the MB has been extensively characterized [63].

Olfactory receptor neurons (ORNs), which are first-order neurons, are situated within the sensilla on the antennae lobe (AL) and maxillary palps, where they detect odors from the outside environment [62,64]. ORNs express specific odorant receptors (ORs) that confer precise olfactory tuning properties [65]. Upon the binding of odorants to ORs, a signal transduction cascade is initiated within the ORNs, leading to the depolarization of activated neurons. ORNs respond to odors by generating distinct sequences of action potentials that encode information regarding the quality, quantity, and duration of odor stimuli [66]. In *Drosophila*, approximately 1300 ORNs are located in the antennae and maxillary palps, with most ORN classes expressing a single OR and projecting their axons onto approximately 50 glomerular targets. Specifically, about 50 classes of ORNs send their axons to corresponding glomeruli within the antennal lobe [62,64,67,68,69,70,71,72,73].

ORNs extend their axons to the antennal lobe, where they establish synaptic connections within glomeruli with projection neurons (PNs) and local interneurons that are either excitatory or inhibitory and thus play a role in the local processing of olfactory information [34]. Glomeruli are densely packed spherical structures within the neuropil that aid in the interpretation of the signals received from the ORNs. Each odorant activates a distinct set of glomeruli, resulting in a conserved glomerular activation pattern in the antennal lobes of flies [74,75]. Initial investigations have indicated that odor stimulation results in similar activation patterns in both the ORN axons and dendrites of projection neurons by expressing calcium indicators in these neural populations [75]. Consequently, sensory information is believed to be faithfully transmitted from primary to secondary neurons, with each odor represented by a unique and sparse spatial activity pattern of sensory or projection neurons.

Uniglomerular projection neurons transmit olfactory signals to two distinct brain regions in insects: the calyx of the MB and the lateral horn (LH). The MB, known for its involvement in insect olfactory learning, comprises approximately 2200 Kenyon cells per hemisphere in adult *Drosophila* [76,77,78,79,80,81]. Kenyon cells exhibit a unique organization, with cell bodies situated in the dorsal posterior region of the brain surrounding their primary dendritic extensions into a neuropil area known as the calyx. The axons of Kenyon cells form a bundle called the peduncle, which subsequently divides into two major branches, one projecting horizontally and the other projecting vertically. The horizontal branch further divides into three lobes, the β, β′, and γ lobes, while the vertical branch consists of the α and α′ lobes. The MB structure encompasses at least three distinct types of Kenyon cells, each projecting axons to specific lobes (αβ, α′β′, and γ), reflecting both developmental and functional differences [82,83,84,85,86].

Numerous studies utilizing whole cell recordings or calcium imaging to examine the representation of olfactory information in MB neurons post-odor exposure have highlighted the significance of cAMP signaling within the MB in early olfactory memory formation [78,84,87,88,89,90,91]. Other studies have shown a transition from dense spatiotemporal population code in the antennal lobe to sparse code in the MB for odor encoding [92,93,94,95,96,97,98,99]. A recent study further proposed that the shift from dense to sparse odor representation in the insect olfactory system involves mechanisms such as spike frequency adaptation at the cellular level and lateral inhibition at the circuit level [100]. The information processed in the MB is relayed to the downstream brain regions through MB output neurons (MBONs) to influence behavior. These MBONs can be cholinergic, glutamatergic, or GABAergic [8,101,102,103,104].

Additionally, PNs transmit olfactory information to the LH, which comprises three types of neurons: LH input neurons (LHINs), LH local neurons (LHLNs), and LH output neurons (LHONs) [1,105,106]. The LH integrates olfactory cues to assign valence (attraction or aversion behavior) to significant odors while also receiving signals from projection neurons to evoke innate behavioral responses [105,106]. In cases where insects encounter unfamiliar odors, signals are sent to the Kenyon cells of the MB for further processing [107,108,109,110,111]. The interaction between the MB and LH thus facilitates the coordination of learned and innate behavioral responses, enabling insects to adapt to their behaviors based on past experiences and innate preferences (Figure 2).

## 4. Olfactory Associative Reward Memories

Classical olfactory conditioning involves the association between a specific odor and an aversive or appetitive stimulus. This type of conditioning necessitates the involvement of specific molecules capable of integrating sensory information from the odorant and aversive or appetitive stimuli within brain regions where the neural pathways for both stimuli are integrated. The pathway responsible for transmitting olfactory information from the antennal lobes to the MB was discussed above (Figure 2). The capacity to detect and react to reward stimuli is a significant evolutionary benefit for animals in their ecological surroundings. The neural pathways involved in these responses are intricate, making pinpointing the core principles governing reward organization and operations challenging. While researchers have gained initial insights into the mechanisms of rewards, this section focuses on the neural circuits that play a role in encoding motivational states, as well as the processes involved in creating memories linked to appetitive rewards.

### 4.1. Neural Circuits Encoding Motivational States

The MB comprises Kenyon cells and incorporates internal-state signals to regulate both learned and innate odor valences [112,113]. When a specific subset of projection neurons is stimulated by an odor, they subsequently activate only a small number of Kenyon cells, with specific cells being activated by the particular odor differing among individual flies [90,114]. This random connectivity indicates that Kenyon cells are unable to represent innate odor valences. Valences can be attributed to Kenyon cells activated by conditioned odors only after the odors are associated with reward or punishment during the learning process. Despite the sophistication of random connectivity, recent studies have shown that it is not entirely accurate. Analyses of the fly brain connectome using electron microscopy have shown bias in the connectivity of projection neurons to Kenyon cells, with projection neurons responsive to food exhibiting a preference for forming synapses with specific subtypes of Kenyon cells [115,116]. These studies indicate that the MB is capable of encoding the valences of certain ethologically significant odors.

Further studies have shown that removing the MB or inhibiting synaptic release from Kenyon cells in flies does not influence their ability to avoid unpleasant odors [77]. However, recent studies have demonstrated that inhibiting specific MBONs that receive signals from Kenyon cells can diminish a fly’s odor avoidance capability even to approach unpleasant odors [117,118,119]. Certain MBON types can elicit either approach or avoidance responses, indicating that they encode positive or negative valences, respectively [101]. Each MBON type targets distinct regions within the MB, with the proposal that all Kenyon cells connect to both positive- and negative-valence MBONs to maintain a balanced output that counteracts one another [101,104]. Consequently, silencing Kenyon cells did not disrupt this equilibrium, although blocking a specific group of MBONs did, indicating the involvement of the MB in regulating odor-induced behavior in cases of network imbalance between Kenyon cells and MBONs.

In the context of associative olfactory learning, flies exhibit behaviors such as approaching rewarded odors and avoiding punished odors after learning. Studies on adult fly associative learning have primarily used aversive odors, which elicit heightened aversion post-conditioning through punishment. The MB influences the learned responses of flies to these odors [77,82]. Reward and punishment signals, either activatory or inhibitory, trigger specific DANs that project to the MB [120,121,122,123]. Each of the 21 DAN types, in a way similar to the MBONs, extends their axonal terminals to distinct regions within the MB, where they can modulate the local synapses between Kenyon cells and MBONs by releasing dopamine and other cotransmitters [103,112,115,117,124,125,126,127]. Consequently, associative learning induces changes in the MB output network balance, thereby enabling rewarded odors to weakly activate negative-valence MBONs, or strongly stimulate positive-valence MBONs to promote approach behavior. Conversely, punishing odors can lead to avoidance behavior through the activation of different neural pathways [101,104]. Essentially, by modifying MBON outputs, associative learning engages the MB in guiding olfactory responses. Notably, this imbalance renders the learned olfactory responses MB-dependent, irrespective of the innate valence of conditioned odors.

Learning can affect the activity of MB outputs, while internal states such as hunger and thirst can influence the modulation of MB DANs and alter the processing of odor information within the MB network. After associating an odor with a sugar reward, flies exhibit a preference for the conditioned odor only when they are hungry [128]. Starvation further triggers the release of neuropeptide F (NPF), which inhibits the activity of PPL1-γ1pedc DANs, thereby allowing the conditioned odor to activate GABAergic MBON-γ1pedc > α/β neurons more effectively [118,128,129]. This activation results in the inhibition of negative-valence MBONs in the β′2 region, thus promoting odor approach behavior [118]. Similarly, thirst induces flies to seek out olfactory cues associated with water through the activation of the lateral horn leucokinin (LHLK) neurons, which are activated by increased hemolymph osmolarity and release leucokinin to inhibit PPL1-γ2α′1 and PAM-β′2a DAN activity [59,130,131]. Leucokinin serves as a common signal for both hunger and thirst, driving hungry flies to approach food-related odors by activating PAM-β′2p and PAM-β′2m DANs [130]. The interplay between leucokinin and other hunger signals (such as NPF and serotonin) at the level of the DANs explains why hunger specifically motivates flies to seek food-related odors over those related to water [132]. These hunger- and thirst-modulated DANs are believed to regulate learned food- and water-seeking behaviors by influencing the odor responses of MBONs in the MB lobe zones, where the axonal terminals of DANs are located, although further research is needed to confirm this hypothesis.

### 4.2. Water-Reward Memory in Drosophila

The internal state of an animal is closely related to its response to rewards, which influences its motivation to seek rewards [112]. For example, thirsty flies perceive water as a rewarding stimulus [59]. The neural circuits and neurotransmitters involved in processing various rewards commonly exhibit structural similarities [133]. Water, which is crucial for maintaining cellular osmolarity in living organisms, further serves as a reward stimulus for thirsty flies. Water detection in *Drosophila* is facilitated by two primary sensory systems: the gustatory system, which is responsible for sensing liquid water, and the hygrosensory system, which detects humidity in the atmosphere. The gustatory system primarily functions through taste sensilla located on the labellum and legs of the fly, which house specialized sensory neurons that express distinct receptors. Among these receptors is the epithelial sodium channel known as Pickpocket 28 (PPK28), which has been identified as crucial for the gustatory reception of water. PPK28 belongs to the DEG/ENaC family of ion channels and is activated by the presence of water, enabling the fly to perceive and respond to liquid water [131,134,135]. This activation results in the depolarization of sensory neurons, which subsequently transmit signals to the central nervous system, particularly to the MBs, where further processing occurs [135]. In addition to the gustatory system, *Drosophila* utilizes a hygrosensory mechanism to detect environmental moisture. This system is mediated by specific sensory neurons that respond to variations in humidity levels. Neurons within the antennal lobes of the fly are sensitive to humidity, facilitating the fly’s navigation towards areas with elevated moisture content [136,137]. The ability to sense both liquid water and humidity is essential for the fly’s capacity to locate water sources and maintain homeostasis. Research indicates that multimodal sensory pathways converge in the MBs, enabling the fly to associate water with specific environmental cues [138].

Dopamine signaling is a critical component of reward processing in the *Drosophila*, in which dopamine neurons are stimulated by internal conditions, encode emotional values, and play a vital role in the formation of cue-induced appetitive memories (Figure 3A) [139,140,141,142]. The central brain of *Drosophila* houses approximately 280 DANs, of which half project to the MB [143,144,145]. These DANs, situated upstream of the MB, can be categorized into two groups (PAM and PPL1), based on the location of the cell body and the direction of axon entry into the MB [144]. PAM neurons have been particularly implicated in the response to water reward (Figure 3B). Among them, PAM-β′1, PAM-β′2, and PAM-γ4 exhibit activity in reaction to water consumption in both water-satiated and thirsty flies, with PAM-β′1 displaying the most pronounced response in thirsty flies, and PAM-β′2 consistently showing high activity, regardless of the fly’s state [59,131]. PAM-γ4 neurons are essential for the formation of water-reward STM following associations between an olfactory cue and a water reward [59,131]. Flies lacking dopamine 1-like receptors (Dop1R1) show reductions in water consumption, in addition to a diminished proboscis extension reflex to water rewards [146]. Furthermore, the proper expression of the DopR1 receptor in the α′β′ neurons of the MBs is necessary for long-term water-reward memory [131].

The transition from water-reward STM to LTM in *Drosophila* is characterized by a sophisticated interaction among neural circuits, neurotransmitters, and molecular mechanisms. Central to this transition is the neural circuitry associated with the MBs. Empirical studies have demonstrated that distinct subsets of DANs are crucial for signaling related to water-reward STM and LTM. In particular, water-responsive DANs that project to the β′ lobe of the MB are vital for the reinforcement of water-reward LTM, whereas water-responsive DANs that project to the γ lobe are responsible for STM formation [131,147]. This finding implies a degree of specialization within the dopaminergic system that facilitates the encoding of various memory types contingent upon the nature of the reward. On the other hand, both water-reward STM and LTM are dependent on the proper functioning of the osmosensitive ion channel Pickpocket 28 (PPK28) for water perception [131,134]. At the molecular level, the consolidation of water-reward LTM is significantly dependent on protein synthesis and the activation of specific transcription factors, notably the cAMP response element-binding protein (CREB). The activation of CREB initiates the transcription of genes essential for memory consolidation, a process that is modulated by the activity of several signaling pathways, including those involving mitogen-activated protein kinase (MAPK) [148,149]. The necessity for de novo protein synthesis suggests that water-reward STM and LTM are not merely variations in memory duration but represent fundamentally distinct processes that engage different molecular mechanisms [58,148]. The expression of a dCREB2-b repressor, specifically in the MB, during adulthood disrupts water-reward LTM while leaving STM unaffected [131], indicating that water rewards, akin to food or alcohol rewards, are primarily encoded by DANs that innervate neighboring horizontal MB compartments and collaborate to evoke an appropriate behavioral response [147,150,151,152,153,154].

The consolidation of water-reward LTM relies on the activation of specific α′β′-related MBONs, including MBON-γ3β′1, MBON-α′2, and MBON-α′1α′3, following memory acquisition [140]. Experimental interference with serotonin receptors in MBONs through RNA disrupts the consolidation of water-reward LTM [140]. Furthermore, disturbances in serotonin biosynthesis or inhibition of neurotransmitter release during memory consolidation in dorsal paired medial (DPM) neurons trigger water-reward LTM disruptions [140]. The findings from the GFP Reconstitution Across Synaptic Partners (GRASP) technique indicate a functional association between DPM neurons and MBON-γ3β′1, MBON-α′2, and MBON-α′1α′3, indicating that serotonin signaling from DPM neurons is crucial for mediating neurotransmitter release in these MBONs, essential for water-reward LTM consolidation [140].

Conversely, the function of γ and αβ neurons plays a crucial role in the retrieval of water-reward LTM [131]. The involvement of dCREB2 activity in the αβ surface and γ dorsal neuron subgroups within the MB is vital for the establishment of water-reward LTM [148]. Neurotransmission originating from these specific neural subgroups is indispensable for water-reward LTM retrieval, although it is not required for the initial acquisition or consolidation stages [148]. Ultimately, the fruit fly relies on neurotransmission from MBON-γ5β′2a and MBON-α3 neurons to retrieve water-reward LTM, which involves the release of glutamate and acetylcholine, respectively, to facilitate LTM recall (Table 1) [131].

### 4.3. Sucrose-Reward Memory in Drosophila

Sucrose memory is another form of reward memory. The diet of *Drosophila* typically comprises a combination of macronutrients, including carbohydrates, proteins, and fats, and micronutrients, such as vitamins and minerals. *Drosophila* ingests various carbohydrate mixtures, including sucrose, fructose, and glucose found in decaying fruits, along with the proteins derived from yeasts that proliferate on fermenting fruit [158,159,160]. The perception of nutritive sugars is facilitated by a specialized gustatory system that employs a specific set of gustatory receptors (GRs) to recognize sweet substances. The primary receptors implicated in sucrose detection are members of the GR family, notably Gr5a and Gr64f. Gr5a has been particularly linked to the detection of trehalose, whereas Gr64f is essential, in conjunction with other GRs, for the recognition of sucrose, glucose, and maltose [161]. These receptors are expressed in sugar-responsive neurons situated on the labellum, proboscis, and various other body regions, enabling *Drosophila* to effectively perceive and react to the presence of sucrose in their surroundings [161,162]. Upon interaction with sucrose, these gustatory receptors activate sensory neurons, resulting in depolarization and the generation of action potentials. This neural activity is subsequently relayed to the central nervous system, particularly to the MBs, which play a vital role in processing sensory information and forming memories related to food reward [163]. The integration of this sensory input allows *Drosophila* to display feeding behaviors, such as proboscis extension and fluid ingestion, in response to sucrose presence [164]. Sucrose as a reward has been the subject of extensive research in *Drosophila*, resulting in a deeper understanding of the mechanisms underlying the functioning of neural circuits during the encoding of sugar rewards, the subsequent modifications in sugar ingestion, as well as the role of olfactory cues associated with sugar in guiding the fly towards sources of nutritious sugar. In contrast, the rewarding properties of proteins remain poorly understood, despite the fact that a fly’s capacity to detect and consume suitable proteins is crucial for its survival and reproductive success, and that glucose consumption is known to also influence protein intake [165]. CREB is essential not only for the development of LTM associated with water rewards and aversive memory, but also for the establishment of LTM related to sucrose [39].

The transition from sucrose-reward STM to LTM in *Drosophila* represents a well-documented phenomenon characterized by complex neural and molecular mechanisms. Research has demonstrated that the processing of sucrose STM and LTM occurs in parallel within the MBs. Specifically, sucrose-reward STM retrieval is mediated by output from the γ neurons, while LTM retrieval is facilitated by output from the αβ MB neurons. Notably, there is no transfer of sucrose-reward STM information from γ neurons to αβ neurons during the formation of LTM [166]. DANs play a pivotal role in signaling the presence of rewards and are integral to both sucrose-reward STM and LTM formation. The activation of distinct subsets of these neurons is contingent upon the type of memory being established. For example, octopaminergic neurons have been identified as responsible for transmitting signals associated with sweet taste, which are critical for sucrose-reward STM, whereas specific dopaminergic pathways are involved in the consolidation of LTM [147,151]. This observation indicates a functional specialization within the reward circuitry of the *Drosophila* brain. Moreover, the involvement of microRNAs and other genetic factors in the regulation of memory formation has been emphasized. For example, certain microRNAs have been identified as influential in the transition from sucrose-reward STM to LTM, with some being essential for sucrose-reward LTM formation while others may enhance STM [167]. This genetic modulation highlights the intricate nature of memory processes in *Drosophila* and indicates that various molecular pathways may be targeted to improve memory performance.

DANs have been shown to be crucial for encoding sucrose-reward memory (Figure 3A), while emerging evidence has indicated that the two primary types of DANs that project to the MB convey information about opposing valences, with PAM neurons predominantly transmitting signals related to rewards [121,122,168]. Specifically, PAM neurons that innervate the α1, γ4, γ5, and β′2 compartments of the MB are activated by stimuli such as water, sucrose, and alcohol [115,122,147,152,155,169]. This compartmentalization enables the encoding of various types of reward memories, allowing for the simultaneous processing of STM and LTM associated with sugar taste and caloric content, respectively [147,151,158,170,171,172]. The activity of PAM-β′2 regulates the feeding rate of sucrose, as well as the subsequent sensation of satiation, while the signaling from adjacent PAM-γ5 is essential to allow the positive reinforcement of the nutritional value of food and sucrose [147,150,151]. Both neuron types innervate the MBON-γ5β′2a, while the optogenetic stimulation of this MBON is sufficient to facilitate the formation of sucrose-related memories [155]. Notably, the plasticity observed within this compartment was influenced by the flies’ dietary experiences. Flies maintained on a high-sucrose diet exhibit diminished activity of PAM-β′2 in response to sugar, which consequently enhances the activity of γ5β′2a and β′2mp MBONs in response to odors sucrose-associated, ultimately impairing the ability to form associative memories related to sucrose [150,171]. Furthermore, the subsequent activation of PAM-β′2 mitigates overeating in flies consuming high-sucrose diets [150], indicating that PAM-β′2 encodes the appetitive characteristics of sugar and integrates signals pertaining to satiety to modulate sugar-seeking behavior and feeding rates. Additionally, dopamine input into the α1, β′1, β′2, and γ5 MB compartments inhibits the proboscis extension reflex in response to sucrose rewards in hungry flies, indicating that hunger may influence dopaminergic circuits within the MB to enhance reward perception and amplify behavioral responses to rewards [173]. The vertical compartments of the MB are more closely linked to 24-h memory retention than to immediate or 3-h memory. The activation of PAM-α1 upon sucrose exposure triggers excitatory input through the α1 Kenyon cells and the α1 MBON [147,156]. Dopaminergic signaling is generally recognized for its role in arousal, learning, memory, and valence encoding across various reward modalities. The modular organization of dopaminergic circuits within the MB is also crucial for encoding distinct types of rewards and generating the corresponding behavioral responses.

In conjunction with dopamine signaling, the acquisition of olfactory cues linked to sucrose consumption necessitates octopamine (OA) signaling (Figure 3C). Genetic mutations affecting tyramine-β-hydroxylase (Tβh), the enzyme essential for OA synthesis, or the OA receptor located in the MB (OAMBs), can further impair the capacity of *Drosophila* to establish a positive association between odors and sucrose [142,174,175]. These OA neurons concurrently influence Kenyon cells and protocerebral anterior medial (PAM) DANs via OAMBs in Kenyon cells, as well as the octopamine β-2 Receptor (Octβ2R) in γ1pedc DANs, thereby facilitating the signaling of positive reinforcement associated with sucrose consumption [121]. In general, OA appears to play a significant role in modulating the overall state, which in turn affects other neurotransmitter systems that govern reward-related behaviors.

Serotonin has been implicated as a significant factor in the reward mechanisms in the *Drosophila* brain (Figure 3D). Although direct evidence for the encoding of reward responses by serotonin is limited, existing research has indicated its involvement in mediating the influence of internal states on reward perception. The activation of a specific subset of serotoninergic neurons provokes starvation-like feeding behaviors in satiated flies, indicating that such neural activation triggers a hunger response, thereby enhancing the motivational drive for sucrose [132]. Within the subesophageal zone (SEZ), serotoninergic neurons that respond to stimuli with contrasting valences regulate balanced sucrose intake via feedback mechanisms from enteric neurons [157]. Two distinct populations of serotoninergic neurons—those projecting beyond the MB and the DPM neurons that synapse with Kenyon cell axons—have been shown to be essential for the formation of sucrose-related memories [39,132,142,176]. Consequently, serotonin primarily serves to modulate behaviors associated with sucrose consumption. Nevertheless, prevailing evidence has suggested that, akin to OA, serotonin plays a role in state modulation, consequently affecting how other neurotransmitter systems shape reward-related behaviors.

Similar to mammals, neuropeptides play a significant role in modulating reward behavior in *Drosophila*, which exhibits a tendency to optogenetically self-activate neurons that express neuropeptide F (NPF), the *Drosophila* equivalent of neuropeptide Y [177,178]. Activation of NPF neurons has further been shown to enhance the preference for odors associated with sucrose [128]. Conversely, the knockdown of NPF receptors in DANs that encode shock results in a diminished preference for sucrose-associated odors [103,120,128]. Additionally, the knockdown of short NPF (sNPF) in the MB, as well as sNPF receptors on PAM neurons, leads to a reduction in the effectiveness of odor-sugar memory encoding [179,180].

Previous studies have explored the role of hormones in *Drosophila* in regulating sugar reward mechanisms, thus highlighting the significance of diuretic hormone 44 (Dh44) and allatostatin A (AstA) in this process. Dh44, which is analogous to the mammalian stress hormone corticotropin-releasing factor, is released from the central brain neurons and is essential for the detection and consumption of nutritive sugars by interacting with its receptors in specific neurons and gut cells [181,182,183,184]. Binding of Dh44 to its receptor is required for behavioral responses to the nutritional value of sugar [184]. AstA neurons, which are responsive to hunger and thirst, inhibit feeding in food-deprived flies, a response counteracted by NPF neurons [185]. Additionally, AstA influences PAM-γ3, thus reducing their activity after sucrose intake, which is linked to reward and associative learning. PAM-γ3 inactivation mediates reward during sucrose consumption and associative learning [123]. These findings emphasize the complex interactions between hormonal and neurotransmitter systems in modulating reward behaviors; however, the exact mechanisms underlying these interactions are still not fully understood (Table 1).

## 5. Conclusions

*Drosophila* has emerged as a highly valuable model for investigating the complexities of learning and memory processes. Behavioral studies conducted on this organism have further established a foundational framework that has yielded significant insights into brain function. The presence of a reward system in insects has substantial practical implications. The simplicity of the insect model enhances the examination of neural network behavior and facilitates the extrapolation of findings to human brain research. Pleasure and displeasure can be conceptualized as fundamental primary emotions, while the learning capabilities exhibited by insects, including instances of relatively sophisticated learning, could be considered essential components of cognitive function. Consequently, it is plausible to attribute the rudimentary aspects of higher-order brain function to insects.

Numerous behavioral investigations involving *Drosophila* have convincingly demonstrated that the MB network is capable of associative learning; however, the precise mechanisms underlying this process remain unclear. The capacity to differentiate appetitive and aversive stimuli is a critical prerequisite for associative learning, as well as a requirement for reward function. However, this capacity does not necessarily entail the presence of a reward in the same manner as in mammals, including the hedonic aspect. Furthermore, associations may arise from various mechanisms: a straightforward ‘learning’ mechanism at the level of a single neuron, an associative mechanism inherently linked to network properties, or a mechanism that supports associative processes through a secondary system. This secondary system may utilize dopamine or another neurotransmitter, potentially either facilitating or reinforcing associations without serving as a direct mediator of rewards. An alternative mechanism may involve a system that conveys information regarding generalized satiety, satisfaction, and rewards in the context of ‘liking’, which can be conceptualized as an abstract operation. We hypothesize that further exploration of the insect reward function may yield additional evidence regarding the ‘liking’ function, likely based on differences in the associated neurochemical systems compared with those in mammals.

The task of deriving mechanistic insights from an extensive array of genes implicated in memory is complicated, as functionally distinct gene networks operate within functionally distinct neural networks to support behavior. Consequently, researchers are beginning to address several key questions, as follows: which specific cells in the peripheral and central nervous systems are engaged in the particular types of learning and memory?; which proteins within these cells modify synaptic transmission to establish a molecular basis for synaptic plasticity?; how stable are these changes, and are they sufficient to allow them to be recognized as distinct memory phases?; and what intricate alterations occur within the memory network to maintain a crucial balance between the continuity and flexibility of both old and new experiences?

A comprehensive understanding of how experiential factors and internal states influence neuromodulation and neural dynamics is essential to elucidating the mechanisms governing the motivational behaviors related to reward seeking and consumption. Additionally, the availability of numerous established behavioral assays coupled with the robust molecular genetics of *Drosophila* could facilitate the development of models of human cognitive disorders and their pharmacological interventions. Investigating the functional implications of human gene mutations and variants using a *Drosophila* model will enhance our understanding of the molecular and behavioral pathophysiologies associated with neurodevelopmental disorders.

## Figures and Tables

**Figure 1 cells-13-01716-f001:**
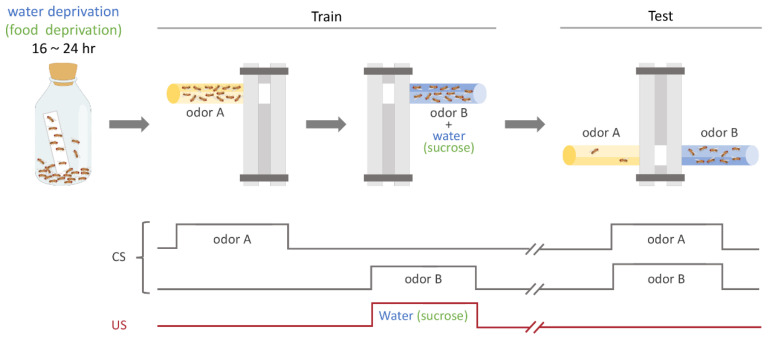
Overview of the associative water- (or sucrose-) reward paradigms. Before the conditioning process, flies were subjected to a period of water (or food) deprivation lasting from 16 to 24 h. The conditioning protocol involved approximately 50 thirsty (or hungry) flies positioned on the elevator of a T-maze. Initially, the flies were transferred to a CS tube, where an odor (odor A: OCT or MCH) was introduced for 2 min. Subsequently, after a 1 min break in exposure to fresh air, the flies were returned to the elevator and guided to the CS+ tube, which contained filter paper soaked in water (or sucrose) and a different odor (odor B: MCH or OCT) for another 2 min period. Memory retention was evaluated at specified intervals after the training session by offering the flies a choice between two distinct odors.

**Figure 2 cells-13-01716-f002:**
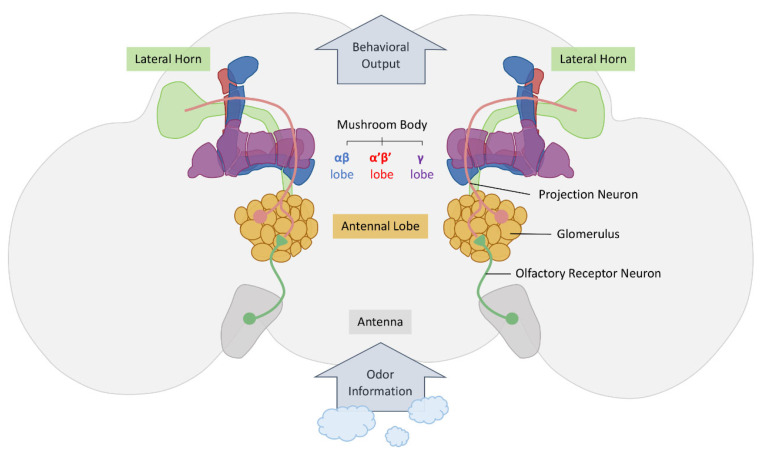
Overview of the *Drosophila* olfactory nervous system. The olfactory nervous system of *Drosophila* is involved in a series of neural processes. Odorants bind to olfactory receptors (ORs) on olfactory receptor neurons (ORNs) located in the sensilla of the antennae lobe (AL) and maxillary palps. The ORNs express specific odorant receptors that facilitate precise olfactory tuning. Subsequently, ORNs send their axons to the antennal lobe, where they form synaptic connections with projection neurons (PNs) in the glomeruli. Uniglomerular PNs transmit olfactory signals to two distinct brain regions in flies: the mushroom bodies (MB) and the lateral horn (LH). Ultimately, this information is relayed to the downstream brain regions through output neurons to influence behavior.

**Figure 3 cells-13-01716-f003:**
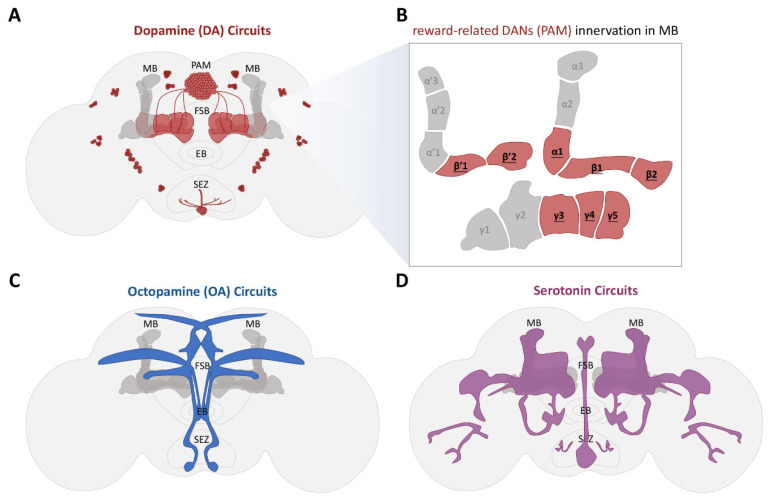
Illustration of the various reward circuits identified in adult *Drosophila*. The identified circuits associated with behavioral responses to rewards encompassed (**A**) dopaminergic circuits; (**B**) the innervation of DANs (PAM) within the MB involved in synaptic connections related to reward across different MB compartments; (**C**) octopaminergic circuits; and (**D**) serotoninergic circuits. (MB: mushroom body; PAM: protocerebral anterior medial; FSB: fan-shaped body; EB: ellipsoid body; SEZ: subesophageal zone).

**Table 1 cells-13-01716-t001:** Water- and sucrose-reward memory circuits.

**Water-Reward Memory**	
Neurons/Molecules	References
PAM-β′1 and PAM-γ4	[59,131]
Dop1R1 receptor in the γ neurons	[59]
Dop1R1 receptor in the α′β′ neurons	[131]
cAMP-responsive element-binding protein (CREB)	[148]
DPM neurons > MBON-γ3β′1, MBON-α′2, and MBON-α′1α′3	[140]
αβ surface and γ dorsal neurons	[148]
MBON-γ5β′2a and MBON-α3	[131]
**Sucrose-Reward Memory**	
Neurons/Molecules	References
cAMP-responsive element-binding protein (CREB)	[39]
PAM-γ5 > MBON-γ5β′2a	[147,155,156]
PAM-α1 > α1 Kenyon cells > MBON-α1	
Octopamine β-2 Receptor (Octβ2R) in γ1pedc DANs	[121]
Serotoninergic neurons	[132,157]

## Data Availability

Not applicable.

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
