# Peer review of "The Neural Correlations of Olfactory Associative Reward Memories in Drosophila"

_cells, 2024, doi:10.3390/cells13201716_

Round 1
Reviewer 1 Report
Comments and Suggestions for Authors
The review “The Neural Correlations of Olfactory Associative Reward 2
Memories in Drosophila” by Yu-Chun Lin, Tony Wu and Chia-Lin Wu reviews our current understanding of learning and memory based on the associative learning models from
Drosophila melanogaster. Though, at times, I found the review a bit confusing and perhaps unnecessarily dense, I still find the review appropriate to be published in Cells. I will list a few general suggestions and some minor issues that I had with the manuscript.
1. One thing that I think about when I read reviews on learning and memory in Drosophila is that is sometimes hard to understand what the authors consider learning and memory, particularly when the authors get into the minutia of molecular/cellular details. As the reader, I follow the neurons from place to place and how they interact but do not get an idea if this this learning or just a circuit diagram of the response to a stimulus. I think it would be helpful if the authors indicate how training/learning influences these events and what the changes from STM to ARM to LTM. This is discussed to some degree later in the manuscript with the sugar and water associated memory, but this could be done throughout or the manuscript could be restructured to generalize the three models examined.
2. As hinted to in point 1, the reader is gets to learn about the circuits mediating the sensing of odors, water and sugar. These are the conditioned stimuli. We get to understand how a CS can interact with the unconditioned stimulus in some way through the MBONs and/or Kenyon Cells. As learning comes from the integration of these two signals, I would like more information on the US circuit and how it effects behavior. The authors do discuss this somewhat in section 4.1, but it is hidden by being mixed in with discussion of other issues. How the CS modifies the US response might be easier to understand conceptually if the authors just clearly defined what happens upon receiving the US. This would then allow the authors to clearly discuss each type of CS network and generalize the mechanisms of learning and memory. Here, I felt the mechanisms of sensing the CS was too much the focus. This made it hard for me to take lessons from each example to understand how learning and memory occur. It was like learning a whole new mechanism each time and that it was up to me to determine the commonalities. While I know that there are differences in how signals get to the point of interaction, one would hope that the mechanisms converge. The organization of the review could make this easier. My suggestion is, of course not the only, or necessarily the best solution, but I am simply giving my feeling as to why I might have had a hard time getting through the review.
3. Sometimes, I found the review included details at points that were not necessary at a particular point or were included in paragraphs where they drew the reader off on a side track that was not pertinent at the time (though later might be useful). The first time I felt this was in the paragraph starting at line 214. From line 225, this could be a new paragraph or even removed from this paragraph and the reader would not lose anything. As is, the reader is trying to integrate too much information without a sufficient frame work in which to place it. At least placing a paragraph separation might help organize it as a separate piece of information to used later. There are other cases of this, but this was the first time that I reacted to it with a written sideline note in my copy.
Minor corrections:
1. Abstract “achieve potential resolution of…”? Why not change this to “resolve”?
2. Line 55, can you add a space after Drosophila?
3. Line 113, change additional (not additive) to something like “different”.
4. Lines 137-139, seems redundant. Can be removed without a loss.
5. Line 175-176, the larval system was explained. They may be quite different. As DANs are also involved in the adult system (as shown later), why not stick to the adult? Also, this reminds me, what about the role of Octopaminergic neurons?
6. Line 288-9, “even leading to unpleasant odors…”, this reads like the diminishment of a fly’s odor avoidance capability can lead them to emit an unpleasant smell.
7. Line 303, “akin” can be interpreted as they are related to MBONs. Perhaps, say “in a way similar to MBONs”.
8. Line 344, repetitive. This is one example. I found this happens in the manuscript where the last sentence of a paragraph often repeats things that do not need repeating. While this can be a good technique to get the readers to remember things, it can get tiresome. I suggest reading the manuscript with idea that things need to be removed, as if there was a word limit.
9. Line 347, "...the fruit fly Drosophila ..." the fruit fly indicates a specific fruit fly but Drosophila is a huge genus. Either use Drosophila alone to represent a generalization or add melanogaster.
10. The authors speak about the cells required for memory retreival without really speaking about what distinguishes this relative to consolidation/ formation at both a conceptual and molecular level. How it is assayed relative to formation and consolidation?
Comments on the Quality of English Language
The English is fine for the most part. I recommended some editing of the organization and tightening up of the text.
Author Response
Reviewer #1
The review “The Neural Correlations of Olfactory Associative Reward 2 Memories in Drosophila” by Yu-Chun Lin, Tony Wu and Chia-Lin Wu reviews our current understanding of learning and memory based on the associative learning models from Drosophila melanogaster. Though, at times, I found the review a bit confusing and perhaps unnecessarily dense, I still find the review appropriate to be published in Cells. I will list a few general suggestions and some minor issues that I had with the manuscript.
- One thing that I think about when I read reviews on learning and memory in Drosophila is that is sometimes hard to understand what the authors consider learning and memory, particularly when the authors get into the minutia of molecular/cellular details. As the reader, I follow the neurons from place to place and how they interact but do not get an idea if this this learning or just a circuit diagram of the response to a stimulus. I think it would be helpful if the authors indicate how training/learning influences these events and what the changes from STM to ARM to LTM. This is discussed to some degree later in the manuscript with the sugar and water associated memory, but this could be done throughout or the manuscript could be restructured to generalize the three models examined.
Author’s responses:
Thank you for your comments. We have incorporated additional paragraphs in Lines 386 to 405 and Lines 462 to 481, respectively, to elucidate the connection between STM and LTM in relation to water and sugar rewards.
- As hinted to in point 1, the reader is gets to learn about the circuits mediating the sensing of odors, water and sugar. These are the conditioned stimuli. We get to understand how a CS can interact with the unconditioned stimulus in some way through the MBONs and/or Kenyon Cells. As learning comes from the integration of these two signals, I would like more information on the US circuit and how it effects behavior. The authors do discuss this somewhat in section 4.1, but it is hidden by being mixed in with discussion of other issues. How the CS modifies the US response might be easier to understand conceptually if the authors just clearly defined what happens upon receiving the US. This would then allow the authors to clearly discuss each type of CS network and generalize the mechanisms of learning and memory. Here, I felt the mechanisms of sensing the CS was too much the focus. This made it hard for me to take lessons from each example to understand how learning and memory occur. It was like learning a whole new mechanism each time and that it was up to me to determine the commonalities. While I know that there are differences in how signals get to the point of interaction, one would hope that the mechanisms converge. The organization of the review could make this easier. My suggestion is, of course not the only, or necessarily the best solution, but I am simply giving my feeling as to why I might have had a hard time getting through the review.
Author’s responses:
Thank you for your comments. We have incorporated an explanation more detailed of the process by which the US signal is received and subsequently transmitted to the MB. In lines 343 to 361, we discuss the mechanisms through which the Drosophila perceives the water signal and how this water reward signal is conveyed to the MB. Additionally, in lines 438 to 451, we elucidate how Drosophiladetects the sugar reward signal and the subsequent transmission of this signal to the MB. The integration of US and CS signals within the MB plays a crucial role in shaping the associative learning behaviors of Drosophila.
- Sometimes, I found the review included details at points that were not necessary at a particular point or were included in paragraphs where they drew the reader off on a side track that was not pertinent at the time (though later might be useful). The first time I felt this was in the paragraph starting at line 214. From line 225, this could be a new paragraph or even removed from this paragraph and the reader would not lose anything. As is, the reader is trying to integrate too much information without a sufficient frame work in which to place it. At least placing a paragraph separation might help organize it as a separate piece of information to used later. There are other cases of this, but this was the first time that I reacted to it with a written sideline note in my copy.
Author’s responses:
Thank you for your comments. We have implemented paragraph separation to enhance organization.
Minor corrections:
- Abstract “achieve potential resolution of…”? Why not change this to “resolve”?
Author’s responses:
We have amended the wording from "achieve potential resolution of..." to "resolve" as suggested.
- Line 55, can you add a space after Drosophila?
Author’s responses:
A space has been inserted prior to the term "Drosophila" as suggested.
- Line 113, change additional (not additive) to something like “different”.
Author’s responses:
The wording has been revised from "additional" to "different" as suggested.
- Lines 137-139, seems redundant. Can be removed without a loss.
Author’s responses:
Thank you for your comments. We have removed the redundant section to avoid any potential confusion.
- Line 175-176, the larval system was explained. They may be quite different. As DANs are also involved in the adult system (as shown later), why not stick to the adult? Also, this reminds me, what about the role of Octopaminergic neurons?
Author’s responses:
We have removed references to the larval system in order to focus exclusively on the adult system as suggested.
- Line 288-9, “even leading to unpleasant odors…”, this reads like the diminishment of a fly’s odor avoidance capability can lead them to emit an unpleasant smell.
Author’s responses:
We have revised our terminology to "even approaching unpleasant odors" as suggested.
- Line 303, “akin” can be interpreted as they are related to MBONs. Perhaps, say “in a way similar to MBONs”.
Author’s responses:
We have revised the phrasing from "akin" to "in a way similar" as suggested.
- Line 344, repetitive. This is one example. I found this happens in the manuscript where the last sentence of a paragraph often repeats things that do not need repeating. While this can be a good technique to get the readers to remember things, it can get tiresome. I suggest reading the manuscript with idea that things need to be removed, as if there was a word limit.
Author’s responses:
Thank you for your comments. The redundant sentence has been removed.
- Line 347, "...the fruit fly Drosophila ..." the fruit fly indicates a specific fruit fly but Drosophila is a huge genus. Either use Drosophila alone to represent a generalization or add melanogaster.
Author’s responses:
We have amended the wording from "...the fruit fly Drosophila ..." to "Drosophila" as suggested.
- The authors speak about the cells required for memory retrieval without really speaking about what distinguishes this relative to consolidation/ formation at both a conceptual and molecular level. How it is assayed relative to formation and consolidation?
Author’s responses:
Thank you for your comments. In the introductory part of section 2.2 (Lines 136 to 143), we incorporated the three stages of memory formation: acquisition, consolidation, and retrieval. Acquisition is the initial phase of memory formation, during which an organism learns to associate a specific stimulus. Consolidation involves the process of enhancing the stability of memories and their resistance to interference. Lastly, retrieval represents the final phase of memory formation, in which the stored memory is accessed and articulated. It is anticipated that this modification will enhance the article's readability.

Reviewer 2 Report
Comments and Suggestions for Authors
Brief summary
Research into the molecular signalling pathways that control learning and memory is crucial for understanding and treating cognitive disorders in humans. Drosophila melanogaster, with its simpler nervous system and genome, allows the study of neuronal plasticity and its genetic basis, which are also relevant to the human brain. In this review the authors focus specifically on olfactory associative reward memories in flies to better understand the underlying neural mechanisms.
General concept comments
This is a well-structured and well-written review that is readily comprehensible. Furthermore, the figures are clearly presented. I have only a few minor comments (see below).
Sepcific comments
Line 55: A typographical error has been identified in the text. The word "Drosophila" is missing a space before the word "as".
Line 102: There is an additional dot preceding the references, which should be removed.
Line 109 and Line 129: It would be reasonable to posit that the bracketed phrase should be interpreted as "or hungry."
Line 127-135: It seems reasonable to conclude that these lines represent the figure legend for Figure 1, as otherwise there would be a repetition. Unfortunately, it is not distinguishable as a figure legend, but it resembles normal text. This may be related to the journal guidelines, but it is worth noting regardless of whether this is the case.
Line 158 and 160: The abbreviation AMP has not yet been introduced and should be explained.
Figure 2: Unfortunately, the figure legend is not distinguishable as a figure legend, but it resembles normal text (see also comment to Figure 1).
Line 302: The abbreviation DANs has already been introduced and used. Therefore, only the abbreviation can be used here.
Line 506. I would recommend writing "Drosphila has emerged...".
Line 530: I would suggest writing "We hypothesize".
Author Response
Reviewer #2
Brief summary
Research into the molecular signaling pathways that control learning and memory is crucial for understanding and treating cognitive disorders in humans. Drosophila melanogaster, with its simpler nervous system and genome, allows the study of neuronal plasticity and its genetic basis, which are also relevant to the human brain. In this review the authors focus specifically on olfactory associative reward memories in flies to better understand the underlying neural mechanisms.
General concept comments
This is a well-structured and well-written review that is readily comprehensible. Furthermore, the figures are clearly presented. I have only a few minor comments (see below).
Specific comments
Line 55: A typographical error has been identified in the text. The word "Drosophila" is missing a space before the word "as".
Author’s responses:
A space has been inserted prior to the term "Drosophila" as suggested.
Line 102: There is an additional dot preceding the references, which should be removed.
Author’s responses:
The extraneous dot preceding the references has been removed as suggested.
Line 109 and Line 129: It would be reasonable to posit that the bracketed phrase should be interpreted as "or hungry."
Author’s responses:
We have made the adjustment from "hunger" to "hungry" as suggested.
Line 127-135: It seems reasonable to conclude that these lines represent the figure legend for Figure 1, as otherwise there would be a repetition. Unfortunately, it is not distinguishable as a figure legend, but it resembles normal text. This may be related to the journal guidelines, but it is worth noting regardless of whether this is the case.
Author’s responses:
We have complied with the journal's guidelines and have adjusted the text size of all figure legends to 9 for enhanced clarity.
Line 158 and 160: The abbreviation AMP has not yet been introduced and should be explained.
Author’s responses:
We have incorporated an explanation of Cyclic Adenosine Monophosphate (cAMP) at line 165 and have utilized the abbreviation at line 167.
Figure 2: Unfortunately, the figure legend is not distinguishable as a figure legend, but it resembles normal text (see also comment to Figure 1).
Author’s responses:
We have adhered to the journal's guidelines and have modified the text size of all figure legends to 9 for improved clarity.
Line 302: The abbreviation DANs has already been introduced and used. Therefore, only the abbreviation can be used here.
Author’s responses:
The term "dopaminergic neurons" has been replaced with the abbreviation "DNAs" as suggedted.
Line 506. I would recommend writing "Drosophila has emerged...".
Author’s responses:
We have revised the phrasing from "Overall, Drosophila have emerged..." to "Drosophila has emerged...".
Line 530: I would suggest writing "We hypothesize".
Author’s responses:
We have made adjustments to the wording, changing it from "We hypothesized..." to "We hypothesize..." as suggested.

Round 2
Reviewer 1 Report
Comments and Suggestions for Authors
The authors have made the changes that I suggested. The manuscript is quite well written so I have nothing more to suggest.
Reviewer 2 Report
Comments and Suggestions for Authors
I would like to express my gratitude for implementing my suggestions and revising the manuscript in accordance with my recommendations. I have no further comments.